# An Investigation on the Relationship between Dust Emission and Air Flow as Well as Particle Size with a Novel Containment Two-Chamber Setup

**DOI:** 10.3390/pharmaceutics16081088

**Published:** 2024-08-20

**Authors:** Steffen Wirth, Martin Schöler, Jonas Brügmann, Claudia S. Leopold

**Affiliations:** 1Department of Pharmaceutical Technology, University of Hamburg, Bundesstr. 45, 20146 Hamburg, Germany; steffen.wirth@uni-hamburg.de; 2Fette Compacting GmbH, Grabauer Straße 24, 21493 Schwarzenbek, Germany; mschoeler@fette-compacting.com (M.S.);

**Keywords:** containment, HPAPI, two-chamber setup, flow barrier, dustiness, dust emissions

## Abstract

In the present study with a novel two-chamber setup (TCS) for dustiness investigations, the relationship between pressure differences as well as air velocities and the resulting dust emissions is investigated. The dust emissions of six particle size fractions of acetaminophen at pressure differences between 0 and 12 Pa are examined. The results show that both simulated and measured air velocities increase with increasing pressure difference. Dust emissions decrease significantly with increasing pressure difference and air velocity. Fine particles cause higher dust emissions than coarse particles. A high goodness of fit is obtained with exponential and quadratic functions to describe the relationship between pressure difference and dust emission, indicating that even moderate increases in pressure may lead to a reduction in the emission. Average air velocities within the TCS simulated with Computational Fluid Dynamics are between 0.09 and 0.37 m/s, whereas those measured experimentally are between 0.09 and 0.41 m/s, both ranges corresponding to the recommended values for effective particle separation in containment systems. These results underline the ability of the novel TCS to control pressure and airflow, which is essential for reliable dust emission measurements and thus provide support for further scientific and industrial applications.

## 1. Introduction

In recent years, highly potent active pharmaceutical ingredients (HPAPIs) have become more prevalent, making the safe handling of these potentially hazardous substances increasingly important. HPAPIs not only cause pronounced pharmacological effects but may also compromise the environment even at very low concentrations [1,2,3,4,5,6]. They are used in various therapeutic areas, e.g., oncology, neurology, endocrinology, and in the treatment of autoimmune diseases. Examples for HPAPIs include cytostatics, hormones, antibody–drug conjugates, and immunomodulators [7,8,9,10,11]. Because of the high health risks associated with HPAPIs, appropriate safety measures, including specific plant and equipment designs, are required. These facilities have to be specifically designed to meet the unique requirements of HPAPI manufacturing, including complex filtration and ventilation systems with high efficiency regarding the removal of fine airborne particles and thus enhancing protection against hazardous airborne substances [12,13,14,15,16]. These precautions include establishing airlocks, maintaining pressure differentials, using high-efficiency filters, restricting access to areas where HPAPIs are handled, as well as the use of specialized personnel-protective equipment to reduce health risks [5,17]. Dust collection systems in milling plants, for example, may also help to minimize airborne exposure and ensure a safer working environment in pharmaceutical manufacturing [18].

A further important issue in this context is to avoid contamination of the environment as well as cross-contamination during production and its accompanying manufacturing processes [3,7,19,20]. As the processing of HPAPIs inevitably generates pharmaceutical dust, appropriate containment systems and special equipment designs are required to minimize potential hazard exposure [1,21,22,23]. In this regard, dustiness describes the tendency of powdery material to generate airborne particles during its handling [21,24,25]. Various processes in pharmaceutical manufacturing, such as material transfer, milling, blending, and granulation are associated with the generation of dust and therefore pose a risk for a potential hazard exposure if HPAPIs are used. Dust generation from pharmaceutical powders is influenced by numerous factors, particularly the physicochemical and mechanical properties of the powders [1,18,26,27,28,29,30,31].

The establishment of occupational exposure limits (OELs), detailed hazard and risk assessments, and a clear understanding and implementation of containment procedures are necessary for a careful approach to handling HPAPIs [3,8,9,10,32,33,34]. The measurement of dust formation is therefore of crucial importance for assessing the health hazards in pharmaceutical facilities. For this purpose, various systems and methods have been developed to reproducibly measure the dust generation of powdery solids. However, most of these systems and methods are not designed for pharmaceutical applications, as the material quantities required for these measurements are relatively high. HPAPIs, for example, are often available only in limited quantities and are relatively expensive [35,36,37,38,39,40]. In this study, therefore, a novel two-chamber model was used to investigate the dustiness of powders under different flow conditions.

The atomization of powders is achieved by applying energy to a powder bulk leading to the distribution of particles in the air. The use of excessive energy may even lead to fragmentation of single particles and therefore to a potentially higher dust generation. Each method for the detection of dust generation involves a different technique of atomization, making a comparison of the different dust detection methods difficult. An exact prediction of dust generation only based on material properties such as particle size and density is not yet possible. Nevertheless, particle size distribution, particle shape, bulk density, residual moisture, as well as cohesive and adhesive forces between powder particles have a significant influence on dust generation [36,38,41,42,43,44].

Reproducible dust generation requires a standardized atomization method with detailed specifications regarding test duration, type and intensity of the mechanical stimulus, and the amount of test material. A high degree of standardization is also essential for performing appropriate dust measurements to ensure reproducibility and a comparability of the results with the conditions during industrial manufacturing processes [38,41,45]. Powdery substances are frequently used to assess the containment performance of equipment by measuring their airborne particle concentration outside of the contained area. The ISPE (International Society for Pharmaceutical Engineering) recommends various surrogates in its Good Practice Guide, including acetaminophen, insulin, mannitol, naproxen sodium, riboflavin, sucrose, and lactose monohydrate [1,46].

Previous studies have shown that Computational Fluid Dynamics (CFD) simulations are suitable for analyzing the flow behavior during dustiness investigations [47,48]. CFD comprises the numerical simulation of fluid dynamics in liquids and gases including dust. By applying numerical models, CFD enables detailed analyses of flow patterns, temperature distributions, and other physical parameters in complex systems. It is an essential tool in various engineering disciplines as well as the pharmaceutical area as it provides profound insights into the behavior of fluids in general. CFD is used to model and optimize a variety of processes, including the characterization of devices and systems for the measurement of airborne particles. The numerical models allow a detailed visualization of the aerodynamics in these devices or systems for investigating the distribution of airborne particles and identify potential areas of particle accumulation [49]. In addition, CFD simulations may illustrate the particle flow in complex systems providing profound knowledge on the atomization process of pharmaceutical powders and allowing the entire production chain to be optimized [25,39,50,51].

One strategy for preventing the unwanted emission of airborne substances is the implementation of a flow barrier, which is described as a displacement concept in EN ISO 14664-4, among others. In this concept, an air flow is directed from the surrounding region into the process zone through positive pressure, effectively preventing the reverse transport of particles. This principle is characterized by the use of a comparatively high air flow accompanied by relatively low pressure differences between the environment and the process zone. In the case of leakage, minimum flow velocities of 0.2 m/s are required for separation of the environment from the process zone. Therefore, flow velocities of 0.4–0.5 m/s are recommended [52,53]. Maintaining a high air flow is necessary to prevent the emission of harmful particles in the event of leakage.

The main objective of this study is to analyze pressure differences within a newly developed two-chamber setup (TCS) by applying these pressure differences to generate both plain diffusive and oppositely directed convective transport of air particles. The surrogate substance, acetaminophen, was atomized inside the TCS and subsequently quantified. In a previous study, it was already demonstrated that different flow conditions within the closed system may be investigated and even small amounts of pharmaceutical powders may be detected [47]. Based on these findings, the present study focuses on the influence of the particle size of the airborne particles and their transport at high air velocities on the resulting dust emissions.

## 2. Materials and Methods

### 2.1. Materials

Acetaminophen (ACAM; Caelo, Hilden, Germany and Fagron, Glinde, Germany) was used as an industry-accepted safe surrogate substance recommended by the ISPE for dustiness. ACAM from Caelo was micronized, whereas that from Fagron showed a particle size of about 20 to 500 µm.

### 2.2. Two-Chamber Setup (TSC)

#### 2.2.1. Design of the TCS

In Figure 1, an illustration of the TCS which was recently described in the literature, is shown [47]. The TCS used in this study consists of 6 mm acrylic glass panes with external dimensions of 618 × 312 × 306 mm. The TCS is divided into two chambers, which are separated by an acrylic glass wall of the same thickness. The two chambers, referred to as the emission and detection chambers, are connected to each other by an orifice with a diameter of 25.4 mm. Consequently, the dimensions of both the emission and detection chambers are 300 × 300 × 300 mm.

The detection chamber has an additional pyramidal construction to improve the detection of airborne particles. By installing this construction, the volume of the detection chamber is reduced to about 15 L, while the volume of the emission chamber remains at 27 L. Both chambers are separated at the top by a removable lid, which is also made of acrylic glass and with dimensions of 630 × 324 × 30 mm. To ensure that the removable lid is sealed, an adhesive elastic rubber seal is applied to the edges of the chambers. By means of toggle locks, which press the lid onto the seals of the chambers to hold them in position, the tightness of the TCS to the environment is ensured. The lid contains two orifices that are positioned centrally above both the emission and detection chambers and are required for atomization and the generation of oppositely directed convective flow. Additional orifices are provided in the lower part of the TCS, being mandatory over the course of the experiment process: In addition to the orifice that establishes a connection to the detection chamber in the emission chamber, in addition, there is another orifice that is necessary for pressure compensation during atomization. Furthermore, there is an orifice for the attachment of a TPE tube (PUN-10X, Festo, Esslingen, Germany), allowing a differential pressure gauge (testo 400, Testo, Titisee-Neustadt, Germany) to be directly connected to the emission chamber. The differential pressure gauge is also connected to an orifice in the detection chamber via a TPE tube. Two type-K thermoelectric couples (thermoelectric couple type-K with TC plug, Testo, Titisee-Neustadt, Germany) are installed in both chambers to be able to monitor the temperature. In addition, the detection chamber contains two diagonally arranged orifices for pressure compensation during evacuation of the chamber. Previous studies have already shown that a diagonal arrangement of the orifices enables a reproducible collection of airborne particles within reasonable time [47,54]. A further orifice is located at the bottom of the detection chamber, towards which the pyramidal construction extends. An IOM sampler (Institute of Occupational Medicine; SKC, Blandford Forum, UK), which is required for the detection of airborne particles, is attached to this orifice. The IOM sampler is further connected to an air sampling pump (AirChek ESSENTIAL Pump, SKC, Blandford Forum, UK).

In Figure 2, a more detailed illustration of the TCS with further constructive parts is shown. The TCS is surrounded by a frame of B-type aluminum profiles to which measuring devices and electronic and pneumatic components for process control are mounted. To adjust predefined pressure differences and to control the pressure difference during the possible transport of airborne particles from the emission to the detection chamber, a differential pressure gauge is required, which is attached to the B-type aluminum profiles. A digital paddle wheel flow meter (35812, ANALYT-MTC Meßtechnik, Mülheim, Germany) is also fastened to the profiles to monitor the set flow rate during the evacuation of the detection chamber. The measurement process is centrally controlled via a programmable logic controller (PLC, Siemens LOGO! 12/24RC, Munich, Germany). The PLC is controlled by a single-board microcontroller (Arduino^®^ Uno Rev3, Ivrea, Italy) and receives a signal from it, which is amplified by a DC/DC converter to provide a sufficiently high input signal. The single-board microcontroller is necessary for synchronizing all devices with the server time and consequently with the measurement phases.

In Figure 3, the system control of the individual pneumatic components by the PLC of the TCS is shown in a piping and instrumentation diagram. The different phases of a measurement are centrally controlled by two 5/2 and two 3/2 solenoid valves. These valves are connected to the corresponding pneumatic components and ensure that the signals sent by the PLC trigger the required operations. This configuration allows for precise control over all phases of the measurement.

In Figure 4, the pneumatic components of the TCS are illustrated. The four output slots of the PLC are connected to two 5/2 and two 3/2 solenoid valves, which are also attached to the aluminum profiles. One of the 5/2 solenoid valves is used for the pneumatic control of the first double-acting ball valve, which is responsible for atomizing the powder samples in the emission chamber. The other 5/2 solenoid valve is applied for the pneumatic control of the second double-acting ball valve (inlet orifice), required for the adjustment of plain diffusion or the generation of an oppositely directed convective flow. The 3/2 solenoid valves represent the outlet orifice and the opening for pressure control, respectively.

Two compressors are used in the design of the TCS. The first compressor (Mega 520-200 D Metabo, Nürtingen, Germany) is required to operate the pneumatic components. By adjusting the flow regulator of the second compressor (Mega 400–50 W; Metabo, Nürtingen, Germany), it is possible to choose between plain diffusive and oppositely directed convective transport. In addition, this 5/2 solenoid valve also pneumatically controls a double-acting cylinder which is connected to the flap required to control the opening or closing of the orifice between the emission and detection chambers.

#### 2.2.2. TCS Measurement Phases

In general, the dustiness measurements with the TCS are divided into three phases, as shown in Figure 5. In the first phase, named the atomization phase, 100 mg of ACAM with different particle sizes are atomized in the emission chamber with the double-acting ball valve at an overpressure of 50,000 Pa for 5 s. The pressure is compensated for the pneumatic control of the 3/2 valve, also for 5 s. During atomization, the double-acting cylinder is actuated so that the flap separates the emission from the detection chamber. This measure ensures that the ACAM particles are homogeneously distributed in the emission chamber without being transferred to the detection chamber.

In the second phase of the measurement, also referred to as the transport phase, the transport of the airborne particles from the emission to the detection chamber is investigated. This phase is intended for analyzing the plain diffusive transport and the oppositely directed convective transport of the airborne particles within the TCS. By adjusting the pressure between the two chambers, it is either feasible to investigate a plain diffusive transport (Δ*p* = 0 Pa) or a convective flow (Δ*p* > 0 Pa). In this phase of the dustiness measurements, the controllable flap is opened for 60 s to allow either diffusive or convective flow between the two chambers and to observe the transport of airborne particles. At the same time, the double-acting ball valve above the detection chamber is opened and the double-acting cylinder is deactivated, thereby connecting the emission with the detection chambers. The pressure difference between the two chambers is verified by the differential pressure gauge.

The third phase, also known as the detection phase, involves the separation of the two chambers by closing the controllable flap after 60 s, thereby quantifying the amount of ACAM transferred from the emission to the detection chamber. Directly after the second measurement phase, the air sampling pump is activated, and the detection chamber is evacuated at a flow rate of 5.0 L/min for 9 min. Meanwhile, pressure compensation is ensured by opening the pneumatic 3/2 valve allowing the flow of filtered air through the diagonally arranged orifices.

#### 2.2.3. Flow Velocities within the TCS

A thermal anemometer (testo 405i, Testo, Titisee-Neustadt, Germany) is used to measure the flow velocities during the transport phase of the dustiness measurements and to compare them with calculated values from CFD (Computational Fluid Dynamics) simulations. In Figure 6, the thermal anemometer is inserted through an acrylic glass plate specifically designed for this purpose. Flow velocities are measured at five different points within the orifice.

In a previous study, pressure differences of 0–4 Pa were achieved by using one compressor and were compared with resulting dust emissions of ACAM with a defined particle size [47]. A second compressor, which is required for a separate pressure cycle, enables higher pressure differences of up to 12 Pa. In addition, ACAM fractions of different particle sizes are used to investigate the influence on dust generation. CFD is also used to investigate the flows resulting from the respective pressure differences within the TCS during the transport phase. Depending on the pressure difference (0–12 Pa), the resulting flow velocities within the TCS are measured. Subsequently, the mean flow velocities determined at the five measurement points in triplicate are calculated. The CFD simulations are carried out with the SimScale software (SimScale, Munich, Germany), applying the k-Omega turbulence model for shear stress transport (k-ω-SST) in a steady-state approach. The initial conditions of the CFD simulations are shown in Table 1.

The flow characteristics for the simulations are shown in Table 2. The boundary conditions are set to a pressure difference between the inlet and outlet in the range of 1–12 Pa in increments of 1 Pa to only take into account the convective flow.

The simulated and measured average air velocities resulting from the pressure differences are used to investigate the effect of air velocity on dust generation. For this purpose, the pressure difference between the two chambers is set to values between 0 and 12 Pa: At a pressure difference of 0 Pa, the plain diffusive transport of ACAM between the detection and emission chambers is investigated, while increasing the pressure difference from 1 to 12 Pa allows the investigation of the diffusive transport of ACAM with an oppositely directed convective flow.

### 2.3. Investigated Powder Fractions

Sieving of the untreated ACAM powders through sieves with mesh sizes of 500, 355, 250, 150, and 63 µm (RETSCH, Haan, Germany) is performed to obtain five different powder fractions, designated as ACAM 2-6. As the ACAM from Caelo was micronized, the sieving procedure was not necessary.

### 2.4. Powder Characterization

#### 2.4.1. True Density

The true density of ACAM is determined with a helium pycnometer (Pycnomatic ACT EVO, Porotec, Hofheim am Taunus, Germany). Between 6 and 8 g of the powder fractions are added to the sample chamber. The true density is determined as the constant value achieved after the samples have reached an equilibrium state. All measurements are carried out in triplicate to minimize the weighing error.

#### 2.4.2. Bulk and Tapped Density

Following monograph 2.9.34 “Bulk and tapped density of powders” of the European Pharmacopoeia [55], the bulk and tapped densities of the different ACAM fractions as well as the micronized ACAM are measured using a vibrating volumeter (STAV 2003, J. Engelsmann, Ludwigshafen, Germany). The 250 mL measuring cylinder is filled with 100 g of ACAM and subjected to tapping. All bulk and tapped densities are measured in triplicate.

#### 2.4.3. Laser Diffractometry

The particle size distribution of ACAM is analyzed by laser diffractometry (Helos KR, Sympatec, Clausthal-Zellerfeld, Germany). A lens with an effective measuring range from 0.5 to 875 µm is used. Compressed air at a pressure of 150,000 Pa is used to disperse the powder samples. The particle size distribution is analyzed using Paqxos software (version 2.0.3, Sympatec, Clausthal-Zellerfeld, Germany). All measurements are carried out in triplicate.

#### 2.4.4. Residual Moisture Content

The residual moisture content of ACAM fraction as well as the micronized ACAM is analyzed by thermogravimetric analysis (TGA; TG 209 F1 Libra^®^, Netzsch-Gerätebau, Selb, Germany). The samples are heated up from room temperature to 105 °C at a heating rate of 10 K/min and are kept at this temperature for 30 min. Measurements are carried out in triplicate.

### 2.5. HPLC Analysis

The quantification of ACAM collected in the glass microfiber filters is performed with a VWR-Hitachi Chromaster 5000 (VWR International, Radnor, PA, USA), equipped with a 250 × 4 mm column (LiChroCART^®^ 250-4, Merck, Darmstadt, Germany) containing an RP-18e phase (particle size 5 µm), as described in a previous study [47]. Briefly, the amount of ACAM in each glass microfiber filter is extracted with 2 mL of the mobile phase (acetonitrile:water (75:25 *v*/*v*)) by shaking with a shaker (Unimax 1010, Heidolph Instruments, Kelheim, Germany). A sample volume of 20 µL is injected into the chromatograph and ACAM content is determined at 245 nm (Chromaster 5430 Diode Array Detector, Hitachi, Chiyoda, Japan). The concentration of ACAM in the sample solutions is linear in the calibration range between 0.002 µg/mL and 2.212 µg/mL (R^2^ = 0.999). The high-performance liquid chromatography (HPLC) assay exhibited reliable analytical performance, with the limit of detection (LOD) determined to be 0.0064 µg/L, and the limit of quantification (LOQ) was at 0.0214 µg/L.

## 3. Results and Discussion

### 3.1. Air Flow Conditions within the TCS

For investigating the air flow conditions within the TCS, it is essential to obtain information about the pressure differences during the individual measurement phases. As already mentioned in Section 2.2.2, the measurement of dust emissions is conducted in three phases, illustrated in Figure 7: the atomization phase, the transport phase, and the detection phase.

In Figure 7a, an overview of all three phases, shown in more detail in Figure 7b–d, is given. This figure clearly illustrates that the TCS operates satisfactorily. In the atomization phase (first phase), the powder in the ball valve is atomized by overpressure. For a time period of 5 s, the pressure in the emission chamber is higher than that in the detection chamber. During this phase, both chambers are separated from each other by the controllable flap. After this time period, the pressure differences decrease to values between 0 and 12 Pa. During this transport phase (second phase), preset pressure differences are reached by opening the ball valve above the detection chamber. After 60 s, the detection phase (third phase) starts. The increase in the pressure differences in this phase is caused by activation of the air sampling pump for 540 s.

The mean values of the pressure differences between the emission and detection chambers during the atomization phase are shown in more detail in Figure 7b. To ensure reproducible and comparable results, the pre-adjusted overpressure is identical for all measurements. With regard to the results, it is important to consider that the atomization phase lasts only for 5 s, and the sampling rate of the differential pressure device is 1/s. The pressure required for atomizing the powder within the emission chamber increases quickly during the 5 s to reach a constant overpressure.

The mean values of the pressure differences between the two chambers during the transport phase in Figure 7c are shown. It is noticeable that the pressure differences, preset between 0 and 12 Pa in clearly distinguished 1 Pa increments, remain constant over the entire duration of this measurement phase because of the second compressor (see Section 2.2.1), despite the permanent opening of the controllable flap during this phase.

In Figure 7d, the detection phase is shown, which is designated by reaching constant pressure difference in a range between 36.1 and 36.7 Pa after a few seconds. This phase ends after exactly 540 s completing the dust analysis. All dustiness measurements last for 605 s.

Overall, the presented results confirm that the TCS enables precise and reproducible control of the pressure differences providing reliable information on the air flow conditions. This is crucial for the consistent measurement of dust emissions and provides a reliable base for further dustiness investigations in the context of containment.

### 3.2. Air Velocities Resulting from Different Pressure Differences

The Computational Fluid Dynamics (CFD) simulations were already carried out in a previous study within a pressure difference range between 1 and 4 Pa [47]. By constructive changes in the TCS and the addition of a second compressor, pressure differences of up to 12 Pa may be reached. The simulations of the resulting average air velocities at pressure differences between 1 and 12 Pa are shown in Figure 8. In this context, it has to be mentioned that pressure difference and air velocity are related to each other.

In Figure 9, a comparison between the simulated and measured average air velocities is presented as a function of the pressure difference. The simulated average air velocities range from 0.09 to 0.37 m/s, while the measured average air velocities are between 0.09 and 0.41 m/s. Again, with increasing pressure difference, both the simulated and measured air velocities increase. However, the measured values are generally slightly higher than the simulated values, especially at higher pressure differences. Despite the slight differences between the simulated and measured data, the curved profiles of both are similar.

### 3.3. Dust Emission Depending on the Pressure Difference/Air Velocity and ACAM Particle Size

In Figure 10, the measured dust emissions at different pressure differences between 0 and 12 Pa are shown. Six different ACAM powder fractions (ACAM 1–ACAM 6) differing in their particle size are examined. The results show that the dust emissions vary depending on the particle size: a significant decrease (*p* < 0.05) in the measured dust emissions is observed with increasing the particle size of ACAM. As expected, the dust emissions also decrease with increasing pressure difference between the emission and detection chambers of the TCS and consequently with higher air velocities. Except for ACAM 6, dust emissions are no longer quantifiable above a pressure difference of 7 Pa (about 0.2 m/s). With the coarsest powder fraction (ACAM 6), dust emissions are measurable only up to a pressure difference of about 4 Pa.

To describe the obtained data series, suitable mathematical functions are used, and their goodness of fit is evaluated. Three different models are examined for this purpose: a linear function, an exponential function, and a quadratic function. The results of the calculations are summarized in Table 3 for the ACAM 1–ACAM 6 samples. The goodness of fit is evaluated by the coefficient of determination R^2^. The first model, a linear function in the form of
y = m⋅x + b,(1)
leads to R^2^ values between 0.674 and 0.882 for the six ACAM fractions, indicating a low goodness of fit. Thus, the linear function does not describe the data appropriately. The second model, an exponential function in the form of
y = a⋅e^−b⋅x^,(2)results in an R^2^ values in the range of 0.926 and 0.987 for the investigated ACAM fractions, providing a considerable goodness of fit. The third model, a quadratic function in the form of
y = a⋅x^2^ + b⋅x + c,(3)
leads to R^2^ values of 0.863–0.977, which also represents a good-fitting model, especially for ACAM 5. Overall, the data are best described by the exponential function, followed by the quadratic function, and the linear function shows the lowest goodness of fit.

It should be noted that these findings are transferable to an only limited extent to the conditions in pharmaceutical manufacturing, as some factors such as the method of atomization, the energy transfer to the powder, the flow conditions, and the composition of the investigated powders have a significant influence on the expected dust emission. These factors may cause a considerable variation in the data, which makes the direct transfer of the experimental results to industrial processes difficult. In addition, the exact configuration and specifications of the TCS play a crucial role in the interpretation of the data and the transferability to other systems and conditions. Overall, the present results illustrate the versatile interactions between particle size, pressure difference, and dust emission.

### 3.4. ACAM Powder Characterization

In the present study, six different ACAM particle size fractions are examined, with ACAM 1 showing the finest and ACAM 6 the coarsest particles. The results of the powder characterization are summarized in Table 4.

The particle size distributions of the ACAM fractions show a distinct difference. The finest ACAM particles (x_10_) show a size of 2.00 µm, while the coarsest particles reach a size of 58.80 µm. The mean particle sizes (x_50_) increase from 9.91 µm for ACAM 1 to 346.16 µm for ACAM 6. The x_90_ values are the lowest for ACAM 1 with 29.98 µm and the highest for ACAM 6 with 574.84 µm.

The true densities of the particle fractions show only slight differences and are similar (about 1.29 g/cm^3^). The bulk and tapped densities of the particle fractions increase from ACAM 1 to ACAM 6. ACAM 1 shows a bulk density of 0.35 g/cm^3^ and a tapped density of 0.57 g/cm^3^, while with ACAM 6, a bulk density of 0.70 g/cm^3^ and a tapped density of 0.82 g/cm^3^ are observed.

The Hausner index, which describes the flow behavior of the powders, decreases from 1.61 for ACAM 1 to 1.16 for ACAM 6. This shows an improvement in flow behavior with increasing particle size. At the same time, the compressibility index decreases from 37.62% for ACAM 1 to 13.84% for ACAM 6.

The residual moisture content is similar in all the samples and ranges between 0.14% and 0.15%.

In summary, the results show that with increasing particle size of the ACAM fractions, the bulk and tapped densities increase, and the flowability improves (decreased Hausner index and compressibility index), whereas the true density and the residual moisture content remain steady.

## 4. Conclusions

To investigate the dustiness of the surrogate substance acetaminophen at six particle size fractions, a novel two-chamber setup is employed. The dust emissions show a significant dependence on the particle size: Fine particles cause higher dust emissions because of their better dispersion within the TCS. Where the correlation between dust emissions and pressure difference or air velocity is concerned, model fitting indicates either an exponential or quadratic relationship. Even moderate increases in pressure difference lead to significant reductions in dust emission. Moreover, there is a significant decrease in dust emission with increasing pressure difference and air velocity.

According to EN ISO 14664-4, one strategy to prevent the undesirable transport of airborne substances is the implementation of a flow barrier. The simulated as well as measured air velocities within the TCS are similar in their range of 0.2–0.5 m/s, recommended in the literature to ensure effective particle separation and control. In fact, depending on the particle size of ACAM, air flow velocities of around 0.2 m/s within the TCS are sufficient to ensure that no quantifiable amounts of ACAM are detected. In summary, the results demonstrate that the investigated TCS enables precise and reliable control of the pressure conditions, which is crucial for reproducible measurements of dust emissions.

In a future study, a more detailed investigation of the flow conditions within the TCS should be performed. The change in the diameter of the orifice between the emission and detection chamber and the resulting change in the cross-section area of the orifice might also influence the flow conditions within the TCS and thus the potential transportation of particles.

## Figures and Tables

**Figure 1 pharmaceutics-16-01088-f001:**
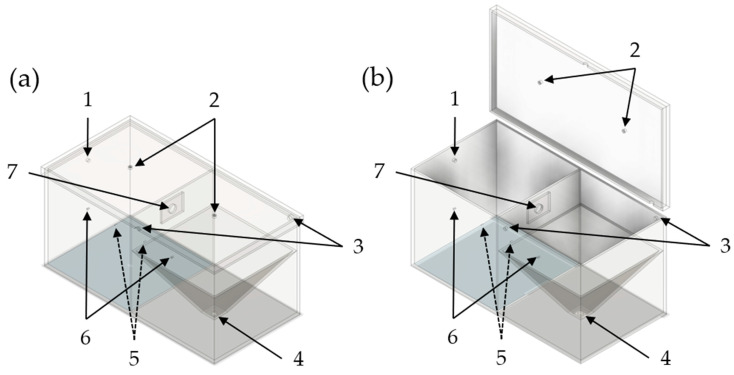
Simplified illustration of the TCS in a closed state (**a**) and an open state (**b**). 1: Orifice for pressure compensation of the emission chamber; 2: Orifices for the double-acting ball valves for atomization and oppositely directed convective flow; 3: Orifices for pressure compensation of the detection chamber during the detection phase; 4: Orifice for the IOM sampler; 5: Orifices for the attachment of thermal sensors of the differential pressure gauge; 6: Orifices for the measurement of the differential pressure with the differential pressure gauge; 7: Connection orifice between the emission and the detection chamber.

**Figure 2 pharmaceutics-16-01088-f002:**
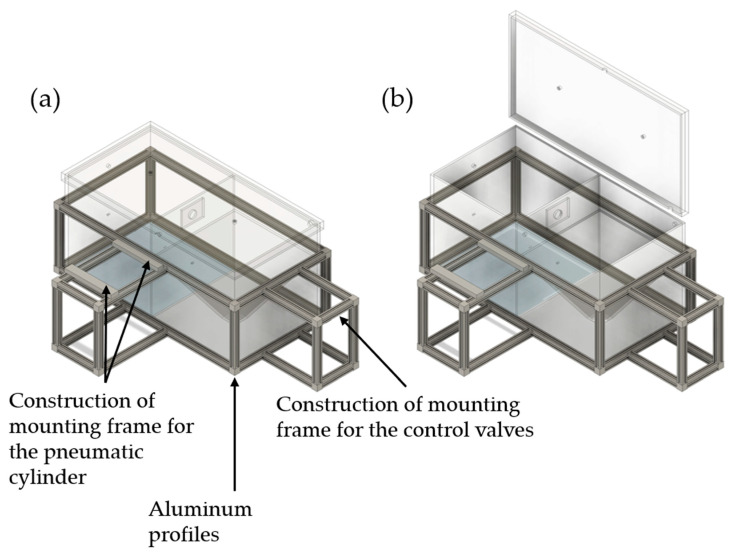
Illustration of the TCS and the constructive components for fastening the pneumatic components: (**a**) TCS in closed state; (**b**) TCS in open state.

**Figure 3 pharmaceutics-16-01088-f003:**
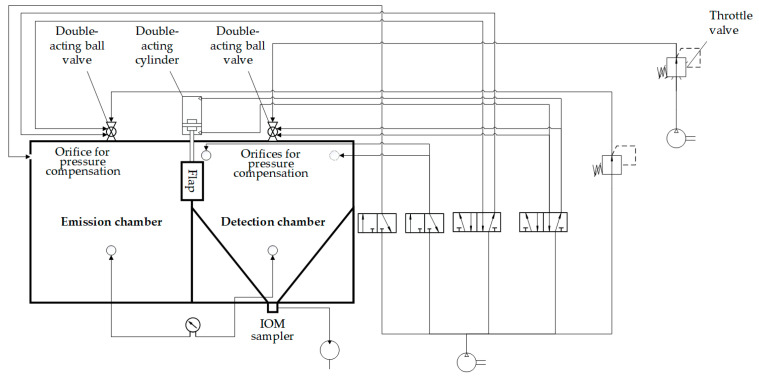
Piping and instrumentation diagram of the individual pneumatic components by the PLC of the TCS.

**Figure 4 pharmaceutics-16-01088-f004:**
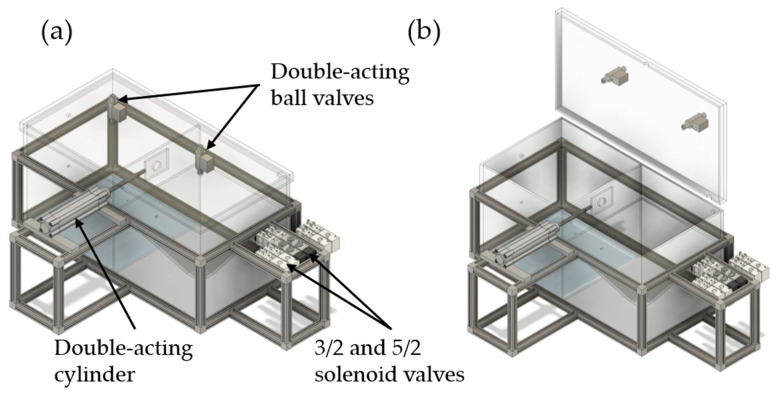
Detailed illustration of the TCS, its aluminum profiles, and pneumatic components: (**a**) TCS in closed state; (**b**) TCS in open state.

**Figure 5 pharmaceutics-16-01088-f005:**
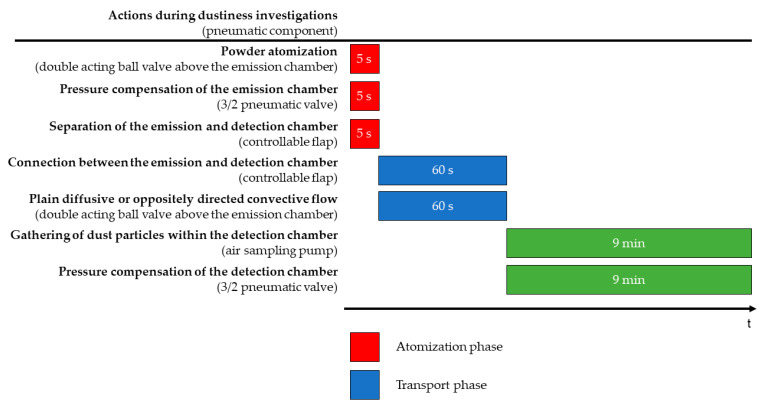
Simplified diagram of the three measurement phases of the TCS.

**Figure 6 pharmaceutics-16-01088-f006:**
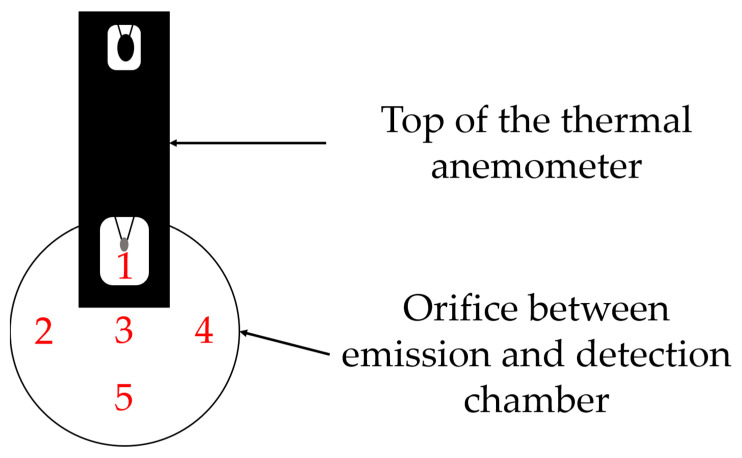
Simplified illustration of the thermal anemometer located within the orifice between the emission and detection chambers featuring five different measurement points.

**Figure 7 pharmaceutics-16-01088-f007:**
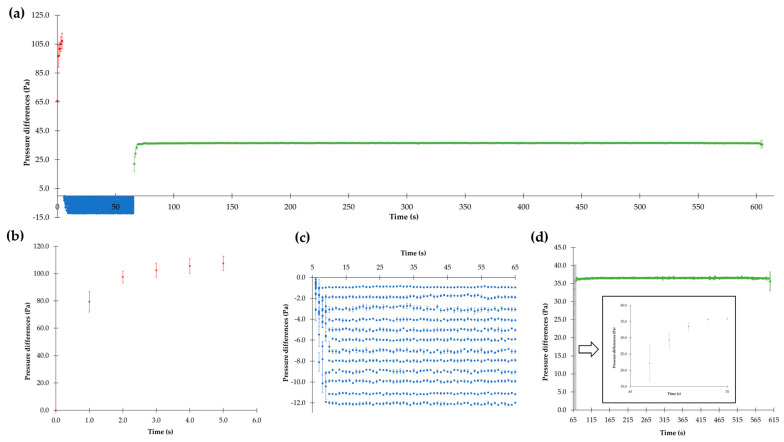
(**a**) Overview of the three measurement phases (red: atomization phase, blue: transport phase, green: detection phase); (**b**) pressure differences during the atomization phase; (**c**) pressure differences of 0–12 Pa during the transport phase; (**d**) pressure differences during the detection phase (means ± SD, *n* = 3).

**Figure 8 pharmaceutics-16-01088-f008:**
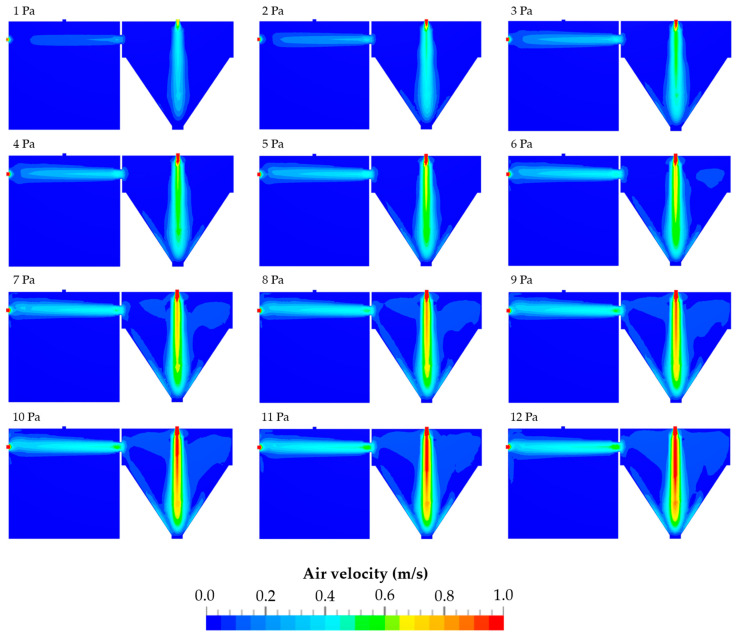
Air velocities at pressure differences of 1–12 Pa between the emission chamber (**left**) and the detection chambers (**right**).

**Figure 9 pharmaceutics-16-01088-f009:**
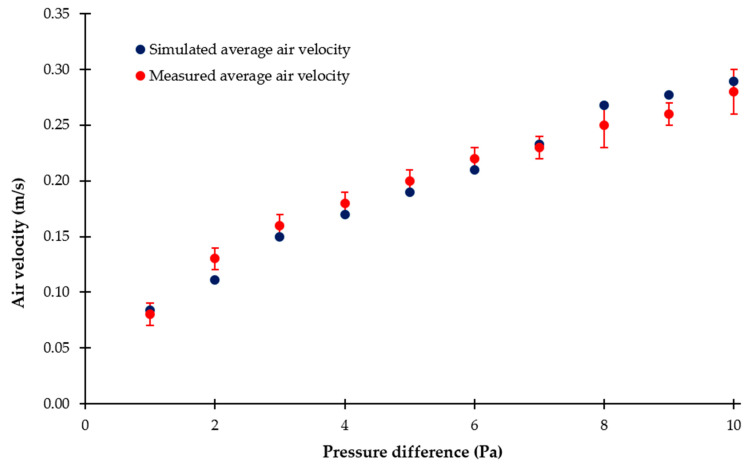
Air velocity versus pressure difference (means ± SD, *n* = 3).

**Figure 10 pharmaceutics-16-01088-f010:**
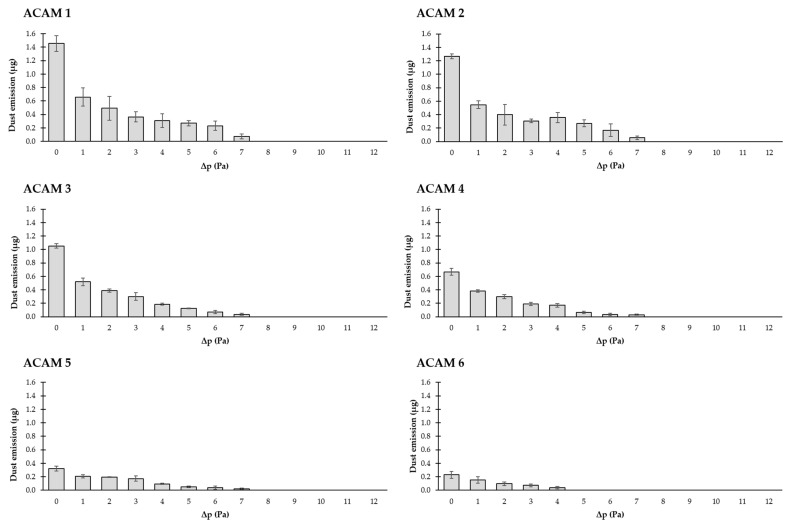
Dust emissions of ACAM 1–ACAM 6 at pressure differences between 0 and 12 Pa (means ± SD, *n* = 3).

**Table 1 pharmaceutics-16-01088-t001:** Initial conditions for CFD simulations.

Properties	Values
Kinematic viscosity	1.529 × 10^−5^ m^2^/s
Density	1.196 kg/m^3^

**Table 2 pharmaceutics-16-01088-t002:** Boundary conditions for CFD simulations.

Properties	Values
Gauge pressure	0 Pa
Δp between inlet and outlet orifice	1–12 Pa in 1 Pa increments
Turbulence kinetic energy [k]	1.297 × 10^−2^ m^2^/s^2^
Specific dissipation rate [ω]	12.49 s^−1^

**Table 3 pharmaceutics-16-01088-t003:** Results of the linear, exponential, and quadratic functions for mathematical modeling for the dust emissions depending on the pressure difference (ACAM 1–ACAM 6).

Sample	Linear Parameters	Linear R^2^	Exponential Parameters	Exponential R^2^	Quadratic Parameters	Quadratic R^2^
ACAM 1	m = −0.087 Pa^−1^ b = 0.820 µg	0.691	a = 1.34 µg b = 0.444 Pa^−1^	0.950	a = 0.014 µg Pa^−2^ b = −0.253 µg Pa^−1^ c = 1.124 µg	0.8804
ACAM 2	m = −0.076 Pa^−1^ b = 0.714 µg	0.689	a = 1.15 µg b = 0.431 Pa^−1^	0.926	a = 0.012 µg Pa^−2^ b = −0.214 µg Pa^−1^ c = 0.967 µg	0.8630
ACAM 3	m = −0.065 Pa^−1^ b = 0.595 µg	0.683	a = 1.00 µg b = 0.472 Pa^−1^	0.982	a = 0.012 µg Pa^−2^ b = −0.204 µg Pa^−1^ c = 0.850 µg	0.923
ACAM 4	m = −0.044 Pa^−1^ b = 0.406 µg	0.724	a = 0.65 µg b = 0.423 Pa^−1^	0.987	a = 0.008 µg Pa^−2^ b = −0.135 µg Pa^−1^ c = 0.573 µg	0.957
ACAM 5	m = −0.025 Pa^−1^ b = 0.233 µg	0.822	a = 0.324 µg b = 0.321 Pa^−1^	0.958	a = 0.003 µg Pa^−2^ b = −0.0633 µg Pa^−1^ c = 0.304 µg	0.977
ACAM 6	m = −0.016 Pa^−1^ b = 0.138 µg	0.673	a = 0.234 µg b = 0.465 Pa^−1^	0.982	a = 0.003 µg Pa^−2^ b = −0.052 µg Pa^−1^ c = 0.206 µg	0.963

**Table 4 pharmaceutics-16-01088-t004:** Overview of the ACAM powder properties.

Properties	ACAM 1	ACAM 2	ACAM 3	ACAM 4	ACAM 5	ACAM 6
x_10_ (µm)	2.00 ± 0.03	4.87 ± 0.59	11.70 ± 0.02	20.10 ± 1.27	27.98 ± 1.00	58.80 ± 5.49
x_50_ (µm)	9.91 ± 0.21	21.99 ± 1.82	70.28 ± 0.07	126.85 ± 6.83	218.32 ± 4.02	346.16 ± 7.13
x_90_ (µm)	29.98 ± 1.78	61.39 ± 3.62	139.24 ± 0.20	250.43 ± 1.64	366.13 ± 2.77	574.84 ± 1.75
True density (g/cm^3^)	1.300 ± 0.009	1.295 ± 0.002	1.299 ± 0.002	1.289 ± 0.006	1.288 ± 0.005	1.288 ± 0.006
Bulk density (g/cm^3^)	0.35 ± 0.02	0.46 ± 0.00	0.53 ± 0.00	0.65 ± 0.00	0.67 ± 0.01	0.70 ± 0.00
Tapped density (g/cm^3^)	0.57 ± 0.01	0.69 ± 0.00	0.76 ± 0.01	0.78 ± 0.00	0.78 ± 0.01	0.82 ± 0.01
Hausner ratio	1.61 ± 0.08	1.52 ± 0.01	1.43 ± 0.00	1.20 ± 0.00	1.16 ± 0.00	1.16 ± 0.01
Compressibility index (%)	37.62 ± 3.20	34.09 ± 0.27	29.88 ± 0.21	16.57 ± 0.17	13.95 ± 0.00	13.84 ± 1.09
Residual moisture content (%)	0.14 ± 0.02	0.14 ± 0.01	0.15 ± 0.02	0.14 ± 0.02	0.14 ± 0.03	0.15 ± 0.04

## Data Availability

Data is contained within the article.

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
