# Peer review of "An Investigation on the Relationship between Dust Emission and Air Flow as Well as Particle Size with a Novel Containment Two-Chamber Setup"

_pharmaceutics, 2024, doi:10.3390/pharmaceutics16081088_

Round 1

Reviewer 1 Report

Comments and Suggestions for Authors

The author reported a new two-chamber setup for dustiness investigations, which can be used to explore the relationship between pressure differences, air velocities and dust emissions, and discussed the dust emission of acetaminophen at six particle sizes under pressure differences conditions ranging from 0 to 12 Pa. It appears that the experiments were carefully carried out. However, the following points need to be clarified in the revised manuscripts before it is considered to be accepted.

1. The introduction takes up 1/6 of the entire piece of writing, but the logic between paragraphs is slightly lacking, making it difficult for the reader to accurately grasp the significance of developing a dual-chamber device. At the same time, the first paragraph lands on preventive measures and lacks proof. It is suggested to add some relevant content, and there are some references suggested to view, such as J. Chem. Health Saf., 2018, 25(1), 36-39, Adv. Mater., 2020, 32(29), 2002361, Mater. Des., 2024, 238, 112615, etc.

2. The description of Figure 3 is only one sentence long, which lacks detail compared to other figures, potentially leading to misunderstandings, and the relevant explanations should be expanded appropriately.

3. Some unnecessary errors in the paper should be corrected. For example, the separation of the header from the table in Table 1, which resulting in a break in the continuity of the paper, and the use of two identical images in Figure 10.

4. Background references are not fully updated and incomplete, with only 2 references for 2022 and 2023. To improve the quality of the manuscript, it is recommended that more up-to-date and relevant references be added.

Author Response

Dear reviewer,

we sincerely appreciate your detailed review of our manuscript and your constructive comments. Your feedback has been invaluable, and we have carefully implemented the changes you recommended. We would like to address the individual points below:

  1. Structure of the introduction: We have revised the structure of the introduction by adjusting the paragraphs to make the text more coherent and readable. In addition, the references you recommended have been integrated into the introduction to emphasize the importance of developing a two-chamber system more clearly.
  2. Description of Figure 3: The description of Figure 3 has been extended to avoid potential misunderstandings and to bring it in line with the other figures in the manuscript.
  3. Correction of errors: The suggested changes, including the corrections in Table 1 and the duplication of images in Figure 10, have been fully implemented.
  4. Updating the references: To further enhance the quality of the manuscript, we have added additional recent and relevant references from 2022 to 2024.

We thank you again for your helpful comments and hope that the changes made will further improve the manuscript.

Reviewer 2 Report

Comments and Suggestions for Authors

In the article under review, the authors propose a new type of experimental set-up for testing dustiness in the air environment and the results of the corresponding experiments performed on this installation. Research of this kind is of great importance from the point of view of protecting the environment and the sanitary and hygienic state of the atmosphere.  The aspect of the problem presented in the work and related to dust formed by . highly potent active pharmaceutical ingredients . is especially important for the biological protection of a humans. Therefore, work in this direction should be welcomed and the lights are of public interest.

The manuscript presents a comprehensive review of the literature on the problem under consideration, which confirms the authors' involvement in the problem under discussion and their understanding of the tasks facing this area.

The description of the experimental set-up and measurement procedure is given in a sufficiently complete form, allowing the reproduction of the proposed methodology.

I have only minor comments.

1.       I do not see a necessity to use three fitting equations, all of them being completely empirical.

2.       Table 3. I cannot understand the authors' "love" for the senseless and incorrect use of up to 5 digits in the approximation of experimental data. This only demonstrates an insufficient understanding of measurement statistics and the concept of confidence limits of measurements. Certainly, this should be corrected.

3.       Table 4 is much better in this aspect, except the third and fourth lines.

Conclusion should be shortened as it must not repeat the content of the study and list the results obtained, but only scientifically important consequences from the investiga

Author Response

Dear reviewer,

thank you very much for your thorough review of our manuscript and the valuable comments you have made. We greatly appreciate your time and effort, and we have carefully considered your suggestions to improve our manuscript.

Regarding your first point about the use of three adjustment equations, we would like to note that our aim was to describe the collected data by different mathematical functions to verify whether these functions can adequately represent the data. We observed that there are mathematical relationships between the dust emission and the pressure difference that may be described by different models. This analysis was important to better understand and model the relationship.

In response to your further comments on the design of the tables and the precision of the values shown, we have revised these sections and adjusted the number of decimal digits to reflect the uncertainties. This makes the presentation clearer and better reflects the statistical accuracy of the results.

With regard to the conclusions, we have reduced repetition and concentrated more on the scientifically relevant consequences of the study to make the summary more concise and clearer.

Thank you once again for your constructive feedback. We are convinced that your comments have contributed significantly to improving the quality of our manuscript.